# Quinazoline Derivative kzl052 Suppresses Prostate Cancer by Targeting WRN Helicase to Stabilize DNA Replication Forks

**DOI:** 10.3390/ijms26136093

**Published:** 2025-06-25

**Authors:** Jia Yu, Gang Yu, Sha Cheng, Liangliang Hu, Ningning Zan, Bixue Xu, Ying Cao, Heng Luo

**Affiliations:** 1Medical College, Guizhou University, Guiyang 550025, China; yu_jia@gmc.edu.cn; 2State Key Laboratory of Discovery and Utilization of Functional Components in Traditional Chinese Medicine, Guizhou Medical University, Guiyang 550014, China; ygfanpuguizhen@163.com (G.Y.); chengsha@gmc.edu.cn (S.C.); 15285034296@163.com (L.H.); zan11658@163.com (N.Z.); bixue_xu@126.com (B.X.); 3Natural Products Research Center of Guizhou Province, Guiyang 550014, China

**Keywords:** WRN helicase, prostate cancer, DNA replication fork, quinazoline derivative

## Abstract

WRN helicases play a key role in DNA replication, repair, and other processes in a variety of tumors. It has become one of the hot targets of genotoxic drugs, but the effect and mechanism of targeting WRN against prostate cancer is still unclear. In our previous study, we found a quinazoline compound kzl052, which has a WRN-dependent inhibitory effect on prostate cancer cells, but its molecular mechanism needs to be further explored. In this study, kzl052 significantly inhibited the growth of PC3 (IC_50_ = 0.39 ± 0.01 μM) and LNCaP (IC_50_ = 0.11 ± 0.01 μM) cells in vitro and showed a good inhibition effect on PCa in vivo. It inhibits PC3 cell growth by binding to WRN proteins and affecting its non-enzymatic function. Then the mechanism of kzl052 against prostate cancer progression was revealed to be by regulating the stability of DNA replication forks and the RB pathway. This study will provide a theoretical basis and treatment strategy for targeting WRN helicase in the treatment of prostate cancer.

## 1. Introduction

Prostate cancer (PCa) is a common malignant tumor of the male reproductive system, which is considered as one of the urgent medical problems to be solved in the male population. Epidemiological statistics show that PCa affects more than 1,400,000 (>7%) people and causes more than 375,000 (3.8%) deaths [1]. The incidence of PCa in Asian countries such as China, Japan, and India is lower than that in Europe and the United States [2]. However, with the change of lifestyle, there has been a significant upward trend in recent years [3]. Most PCa patients are diagnosed at the middle or advanced stage. Although androgen deprivation therapy (ADT) can inhibit tumor growth, the vast majority of PCa patients will relapse. And it develops into castratie-resistant prostate cancer (CRPC), which has a poor prognosis with a median survival of only 12 months [4]. Abnormalities of the androgen receptor (AR) and related pathways remain one of the most important mechanisms for the progression of CRPC [5]. Therefore, its treatment mainly includes AR antagonists, targeted radiotherapy, etc., but the effect is still not satisfactory, and new mechanisms of action and targets need to be explored [6]. Another effective strategy is to target genomic instability. Classic PARP inhibitors (such as olaparib, etc.) have all been approved for the treatment of metastatic CRPC with homologous recombination (HR) deficiency and have shown good clinical effects [7]. However, PARP inhibitors also have significant problems. The first one is that patients with BRCA1/2 mutations only respond partially, and the patient coverage is relatively small [8]. Second, with the prolongation of the treatment process and the change of the pathogenic mechanism, the acquired resistance eventually leads to the failure of the treatment [9]. WRN helicase (WRN) is a key factor that regulates DNA replication, transcription and repair, etc. And it plays an important role in aspects such as the stability of the genome and replication forks [10], but there are relatively few studies on its targeted therapy for CRPC.

WRN is an enzyme encoded by the WRN gene, which belongs to the RecQ DNA helicase family and can have both DNA helicase and exonuclease activities [11,12]. WRN regulates DNA repair, replication, telomere maintenance, and other functions by interacting with proteins related to nucleic acid metabolism [13,14]. WRN gene defects can lead to Werner syndrome (WS), which is an extremely rare human recessive genetic disease characterized by highly unstable genome and chromosomal abnormalities and is associated with premature aging and a variety of cancers [15,16]. Studies have shown that WRN helicase is abnormally increased in various malignant tumor tissues such as breast cancer [17], colorectal cancer [18], and leukemia [19], which regulates genomic stability to participate in the occurrence and development of tumors. In prostate cancer, the single nucleotide polymorphism of WRN Leu1074Phe was associated with the risk of prostate cancer in Chinese men, and the TG/GG genotype displayed a decreased prevalence of prostate cancer compared with the TT genotype [20]. WRN G327X mutations create a premature translational stop signal and can cause a category of pathogenicin hereditary prostate cancers [21]. Therefore, targeted inhibition of WRN helicase may be beneficial for the treatment of some prostate cancer patients.

In previous studies, we reported a series of WRN inhibitors with good anti-prostate cancer activity in vivo and in vitro [22,23], but the molecular mechanisms are still unclear. This study aimed to explore the molecular mechanism of the quinazoline analog kzl052 targeting WRN against prostate cancer and provide a theoretical basis and treatment strategy for targeting WRN helicase in the treatment of prostate cancer.

## 2. Results

### 2.1. kzl052 Inhibits the Growth of Prostate Cancer Cells In Vitro

MTT results showed that kzl052 significantly inhibited the growth of PC3 (IC_50_ = 0.39 ± 0.01 μM) and LNCaP (IC_50_ = 0.11 ± 0.01 μM) cells (Figure 1A–C). Besides, the results of normal cytotoxicity showed that the IC_50_ of kzl052 in normal human liver cells LX2 (3.12 ± 0.09 μM) was significantly higher than that of the positive control NSC 617145 (WRN inhibitor) (1.81 ± 0.27 μM), while it was significantly higher in normal human renal epithelial cells HK2 (0.92 ± 0.24 μM) than that of the NSC 617145 (5.64 ± 0.96 μM) (Table 1), indicating that the normal human liver cells toxicity of kzl052 is significantly lower than that of the NSC 617145. Bright field and Hoechst staining showed that kzl052 promoted PC3 and LNCaP cell death (Figure 1D). The above results showed that kzl052 had a good inhibitory effect on PCa cells, and the 100 nM concentration treatment was selected for the following experiment.

### 2.2. The Sensitivity of kzl052 Against Prostate Cancer by Targeting WRN

Previously, we reported a novel WRN inhibitor against PCa in vitro and in vivo, but the correlation between the WRN protein and PCa is unclear [22,23]. The Western blotting results showed that the WRN protein was expressed differently in various prostate cancer cell lines and was highly expressed in 22RV1 and PC3 cells (Appendix A). The analysis results of the UALCAN database showed that there was no significant difference between WRN expression and the PCa patients survival (Appendix A). However, WRN has a certain relationship with the staging and progression of prostate cancer (Appendix A). Besides, the molecular docking results showed that kzl052 was bound to WRN through hydrogen bonds (ARG A:732 and ARG A:857), Halogen, and Pi–cation and Pi–Pi bonds (Figure 2A), and the binding energy was −6.5 kcal/mol. And the CETSA results confirmed that kzl052 was bound to the WRN protein (Figure 2B), but kzl052 did not affect the activity of WRN helicase (Appendix A). We further constructed WRN-inhibited PC3 cell lines by lentiviral transfection (Figure 2C). MTT results showed that kzl052 was more insensitive to WRN-silenced PC3 cells (Figure 2D). In addition, the Western blot results showed that kzl052 significantly inhibited WRN protein expression (Figure 2E). The above results suggested that kzl052 may exert an anti-proliferative effect by targeting WRN and its non-enzymatic functions.

### 2.3. kzl052 Promotes DNA Damage in PCa Cells

WRN helicases have been reported to play a key role in genome stability and DNA damage repair [10]. DAPI (4′,6-diaminyl-2-phenylindole) is a fluorescent dye that binds to DNA and is primarily used to label the nucleus or DNA. γ-H2A.X is an important marker of DNA damage response, which mainly marks the location and extent of DNA damage [24]. PC3 and LNCaP cells were treated with kzl052, and DNA damage was detected by immunofluorescence. The results showed that there was no significant change in DAPI staining after treatment with kzl052 for 24 h, while γ-H2A.X staining was significantly increased in both PC3 and LNCaP cells (Figure 3). These results suggested that kzl052 significantly aggravated DNA damage in PC3 and LNCaP cells.

### 2.4. Signaling Pathway Analysis of kzl052 Inhibiting PCa Progression

To further explore the signaling pathways of kzl052 in inhibiting PCa progression, the key targets of kzl052 were predicted by the SwissTargetPrediction and Super-PRED database. Key genes contributing to the progression of CRPC were analyzed by the GeneCards database. Venny results showed that the two aggregators shared 26 key genes (Figure 4A,B). The Gene Ontology (GO) function annotation of these genes showed that kzl052 may regulate the processes of PCa cell proliferation, enzyme binding, and the cell cycle–kinase complex (Figure 4C–E). The pathway analysis showed that it may regulate key pathways in PCa such as RB and cell cycle–replication fork complexes (Figure 4F). These results suggest that kzl052 may regulate PCa progression through cell cycles, DNA replication fork stability, and the Rb pathway.

### 2.5. kzl052 Regulates the Stability of the DNA Replication Fork and RB Pathway

It is reported that PARP1 is involved in DNA repair, and Bax/Bcl-2 is a pair of key genes for promoting/inhibiting apoptosis [25,26], while RB and PTEN mainly regulate one of the signals of the prostate cancer cell cycle, DNA damage response, and tumor suppression [27,28]. And the SRC is one of the key oncogenes in prostate cancer [29]. Western blot results showed that kzl052 significantly up-regulated the expression of PARP1, Bax, PLK1, and RB1 and inhibited the protein levels of Bcl-2, SRC, and PTEN to promote cell apoptosis (Figure 5A,B). In addition, the proliferating cell nuclear antigen (PCNA) is closely related to the synthesis of cellular DNA and is a good indicator reflecting the proliferation status of cells [30]. Replication protein A (RPA) is an important DNA-binding protein in cells and plays a key role in the processes of DNA replication, repair, and recombination [31]. Flap structure-specific endonuclease 1 (FEN1) is mainly involved in the process of DNA repair and replication [32]. And Mre11 plays a key role in cutting incorrect DNA during the process of double-strand break repair in cells [33]. The Western blot results showed that kzl052 significantly inhibited the expression of PCNA, RPA, and FEN1 and up-regulated the protein level of Mre11, suggesting that kzl052 promoted the instability of the DNA replication fork complexes (Figure 5C). It is suggested that kzl052 inhibits the progression of PCa by regulating the stability of DNA replication forks and the RB signaling pathway.

### 2.6. kzl052 Inhibits the Progression of Prostate Cancer by Regulating DNA Replication Fork Stability

The acute toxicity results of kzl052 are shown in Table 2. The results show that the LD50 of kzl052 within 24 h is between 200 mg/kg and 500 mg/kg, indicating a minor toxicity in vivo. Animal experiments showed that kzl052 significantly inhibited the growth of PC3 cells in vivo (Figure 6A–C). H&E staining showed that there was no significant difference after kzl052 treatment compared with the control group (Figure 6D). Furthermore, kzl052 had minimal effects on mouse body weight and internal organ morphology following treatment (Figure 6E,F). And the immunohistochemical results of tumor tissues conformed that kzl052 promoted the instability of DNA replication fork complexes (Figure 6G). These results suggest that kzl052 has a good effect on PCa treatment by regulating DNA replication fork stability in vivo.

## 3. Discussion

WRN helicase is very important for genome stability, and cells with WRN mutations are more prone to DNA damage and breakage [13,34]. In recent years, based on this genomic vulnerability caused by WRN mutations, its targeted inhibitors have achieved surprising synthetic lethal effects in microsatellite instability (MSI) cancers [18,35]. So far, only a few WRN inhibitors have been reported, such as HRO761 [36], NSC19630, or NSC617145 [16], and they are mainly targeted at patients with a MSI-H type. However, the proportion of MSI-H patients in various tumor types, including PCa, is relatively low, and WRN inhibitors have no significant effect on microsatellite stability (MSS) tumors, with limited coverage [8]. Our findings identify a novel WRN inhibitor that effectively inhibits MSS PCa progression with less in vivo toxicity, providing a new option for WRN inhibitors to treat PCa tumors. Compared with other WRN inhibitors, kzl052 cannot be used as a WRN inhibitor to induce synthetic lethality. We think that it may be related to its non-enzymatic mechanism [37].

It has been reported that WRN is essential for the reactivation of disturbed replication forks. A WRN dimer can bind to the DNA replication fork and unwrap it. Under replication stress, WRN binds to the replication fork in tetramer form and is dimerized during replication fork reactivation [38]. In this process, binding to a RPA protein is essential for WRN to restart the replication fork [39]. In addition, WRN can also interact with FEN1 to protect disturbed replication forks from excessive excision by MRE11, thereby reducing replication-stress-induced genomic instability [40]. In this study, it was found that kzl052 significantly down-regulated the key protein expression of the replication fork (PCNA, RPA, and FEN1) and up-regulated Mre11 to cleave damaged DNA, suggesting that kzl052 inhibited the proliferation of PCa cells by affecting the stability of the replication fork (Figure 7). Furthermore, we did not discuss the relationship between the RB1 signal and the stability of the WRN–DNA replication fork, and therefore it was not described in the mechanism diagram (Figure 7).

Interestingly, WRN and BRCA2 co-regulate replication fork stabilization in cancer cells and promote cell proliferation [41,42]. In BRCA2-deficient cells, including prostate cancer, WRN is preferentially localized on the replication fork. WRN inhibitors trap WRN onto the chromatin and promote rapid degradation of unprotected replication forks by MRE11, leading to chromosomal instability [42]. Therefore, inhibition of WRN can enhance the cytotoxicity of PARP inhibitors (olaparib) in BRCA2-deficient ovarian cancer cells, suggesting that WRN may be used as an alternative or auxiliary target for PARP inhibitors [43]. In this study, kzl052 significantly promoted DNA damage and replication fork instability in non-BRCA-deficient PC3 prostate cancer cells [44]. However, the currently reported WRN inhibitors have no significant effect on non-MSI-H tumors. It could be the advantage of kzl052 in non-BRCA-deficient prostate cancer. Furthermore, kzl052 significantly increased the expression of the PARP1 protein, which could be a synergistic role in promoting cell death with PARP inhibitors, but it has not been explored.

In this study, we found that kzl052, a novel WRN inhibitor, significantly inhibited the progression of prostate cancer in vitro and in vivo by regulating replication fork stability. This study provides a new idea for the treatment of advanced prostate cancer.

## 4. Materials and Methods

### 4.1. Cell Culture and Transfection

Human prostate cancer cell lines PC3 and LNCaP cells were stored in the Biology Laboratory of the Natural Products Research Center of Guizhou Province (Guiyang, China). Cells were cultured in DMEM supplemented with 10% fetal bovine serum (BI, Nahariya, Israel) and 1% penicillin and streptomycin (Solarbio, Beijing, China) and incubated at 37 °C under 5% CO_2_, 95% air, and 95% humidity. The pLV-shWRN-GFP-Puro and blank control plasmids were constructed by Chongqing UNIBIO Biotechnology Co., Ltd. (Chongqing, China). The transfection steps were based on previous published studies [16]. The inhibitory effect of the WRN protein in PC3 cells was detected by a Western blot assay.

### 4.2. MTT Assay

Cell suspensions were prepared in a logarithmic growth phase and seeded in 96-well plates at a density of 5000–8000 cells/well for 12 h and treated with different concentrations of kzl052 for 48 h. A 5 mg/mL MTT dye solution (Solarbio, Beijing, China) was added to all cells at 37 °C for 4 h. Then DMSO was added after removing the supernatant, and the crystals were completely dissolved by shaking at 37 °C for 15 min. The absorbance was measured at 490 nm by a microplate reader (Gene, HongKong, China). The cell proliferation inhibition rate was calculated according to the following formula: (1-experimental group OD/control group OD) × 100% with DMSO as a negative control and the WRN inhibitor (NSC 617145, GLPBIO, Montclair, CA, USA) as a positive control.

### 4.3. Hoechst Staining

PC3 and LNCaP cells were cultured and treated with different concentrations of kzl052 for 24 h and then 5 μg/mL of a Hoechst 33258 dye solution (Solarbio, Beijing, China) was added for 5–10 min after PBS washing. The fluorescence microscope (Leica, Wetzlar, Germany) was used for photo observation.

### 4.4. AutoDocking

The molecular docking technique was used to predict the interaction between the compound kzl052 and the WRN protein. Firstly, KingDraw V3.0.1.0 software was used to draw the 2D structure of compound kzl052, and the ChemBio3D Ultra 14 software (V14.0.0.117) was used to convert the 2D structure of kzl052 to its 3D structure, and the minimum free energy conversion was performed. Then, the 3D structure of the WRN protein (PDB: 6YHR) was downloaded from the PDB database “https://www.rcsb.org/ (accessed on 17 October 2022)” as the receptor file. The PyMOL 2.5 (V2.5) software was used to remove water molecules and small molecule ligands. The AutoDockTools (V1.5.7) software was used to add hydrogen ions and determine the active pocket position of the target protein. Finally, molecular docking was performed using the AutoDockVina (V1.2.0) software, and the maximum energy difference (energy_range) was set to 5. The molecular docking results were visualized using PyMOL and the Discovery Studio Client (V4.5) software.

### 4.5. Cellular Thermal Shift Assay (CETSA)

PC3 cells were cultured and treated with 100 nM kzl052 for 2 h. The cells were collected into 1.5 mL EP tubes, treated with a metal bath at different temperatures (43 °C, 46 °C, 49 °C, 52 °C, and 55 °C) for 3 min, and balanced for 2 min at room temperature. The supernatant was collected by centrifugation after three cycles of liquid nitrogen in a 37 °C water bath, and then a SDS-loading buffer and denatured protein was added at 100 °C for 10 min. Finally, the content of the binding of the WRN protein was detected by Western blot.

### 4.6. Bioinformatics Analysis

The relationship between WRN expression and PCa patients’ survival was evaluated by using the UALCAN database “http://ualcan.path.uab.edu/index.html (accessed on 29 May 2025)”. The SwissTargetPrediction “http://swisstargetprediction.ch/ (accessed on 17 January 2022)” and Super-PRED database “https://prediction.charite.de/index.php?site=chemdoodle_search_target (accessed on 22 April 2025)” were used to predict the possible targets of kzl052. The key predicted targets were intersected with the key CRPC genes obtained from the GeneCards tumor database “https://www.genecards.org/ (accessed on 28 March 2023)” through the Venny map “https://www.bic.ac.cn/EVenn/#/ (accessed on 2 April 2025)”. The key possible targets from Venny were further analyzed by Protein-Protein Interaction (PPI) “https://cn.string-db.org/ (accessed on 2 April 2025)”. The key signal pathway of kzl052, anti-CRPC, was obtained by GO analysis and pathway analysis “https://cn.string-db.org/ (accessed on 2 April 2022)”.

### 4.7. Western Blot Assay

The cells treated by kzl052 were collected, washed by precooled PBS, and removed by centrifugation at 1000 g/5 min. A RIPA lysate (Solarbio, Beijing, China) was added for 30 min on ice. The protein concentration was determined by a BCA kit (Solarbio, Beijing, China) and denatured at 105 °C. The target proteins were isolated by SDS-PAGE gel electrophoresis and transferred to the PVDF membrane (Millipore, Bedford, MA, USA). A 5% BSA was added to block for 2 h. Primary antibodies (PNCA, RPA, FEN1, Mre11, PARP1, Bax, Bcl-2, PLK1, RB1, SRC, and PTEN; 1:1000 dilution, HUBIO, Hangzhou, China) were added and incubated at 4 °C overnight. The next day, the PVDF membrane with target proteins were washed by TBST and incubated with an HRP-labeled IgG secondary antibody (1:100,000 dilution, HUBIO, Hangzhou, China) for 2 h. After TBST washed, the target proteins were isolated and analyzed by a chemiluminescence instrument (BioRad, Hercules, CA, USA).

### 4.8. Immunofluorescence Assay

Cell slides were prepared and treated with 100nM kzl052 for 24 h. A 4% PFA was added for fixation after PBS washing, and then the cells were treated with Triton X-100 (Sigma-Aldrich, Saint Louis, MO, USA) for 30 min. The cells were blocked with 4% BSA for 2 h and incubated overnight at 4 °C with a γ-H2A.X primary antibody (1:1000, CST, Boston, MA, USA). A fluorescent secondary antibody (1:50; CST) was added and incubated in the dark for 2 h. After washing with PBS, a 10 μg/mL DAPI staining solution was added and treated the cells for 5–10 min. The expression and localization of γ-H2A.X was detected by inverted fluorescence microscopy (Leica, Wetzlar, Germany).

### 4.9. Animal Expriment

Male BALB/c-null mice (4 weeks and 18–20 g) were purchased from SiPeiFu Biotechnology Co., Ltd. (Beijing, China). All SPF-grade mice were fed in the animal room with an IVC system barrier environment (Natural Products Research Center of Guizhou Province). The ambient temperature was maintained at 22 ± 2 °C, the humidity was 40–60%, and the light–dark cycle was maintained for 12 h to simulate the normal external environment. The mice took food and water autonomously. Xenografts were performed by subcutaneously implanting PC3 cells (1 × 10^7^/0.1 mL). When the tumor size was 100 mm^3^, the mice were randomly divided into 2 groups (*N* = 8). One group was intraperitoneally injected with kzl052 (5 mg/kg), and the other group was given the same dose of the solvent. The body weight and tumor volume was recorded every two days. The tumor volume (mm^3^) = 0.5 × long diameter × short diameter^2^. At the end of the experiment, the mice were euthanized and dissected to measure the weight of tumor tissue, heart, liver, spleen, kidney, and lung.

### 4.10. Hematoxylin and Eosin Staining (H&E)

The tumor tissue sections were placed sequentially in xylene I/10 min; xylene II/10 min; absolute ethanol I/5 min; absolute ethanol II/5 min; 95% ethanol/5 min; 90% ethanol/5 min; 80% ethanol/5 min; and 70% ethanol/5 min. Then the sections were treated in a hematoxylin staining solution (Solarbio, Beijing, China) for 3–5 min. The slices were treated by 1% hydrochloric acid alcohol for 5–10 s. The sections were treated in an eosin staining solution (Solarbio, Beijing, China) for 1–3 min. Next, the slices were dehydrated with 95% ethanol/5 min; anhydrous ethanol I/5 min; anhydrous ethanol/5 min; xylene I/5 min; and xylene II/5 min. Finally, the sections were sealed with a neutral gum (Solarbio, Beijing, China). The H&E staining results were photographed and analyzed under light microscopes (Leica, Wetzlar, Germany).

### 4.11. Immunohistochemical Staining

The tissue sections were roasted, dewaxed, hydrated, antigen repaired, blocked, and incubated with a polyclonal antibody (PNCA, RPA, Mre11, PARP1, and RB1 at a 1:50 dilution, HUBIO, and γ-H2A.X at a 1:50 dilution, CST) at 4 °C overnight. The HRP-labeled IgG secondary antibody (1:1000 dilution, HUBIO, Hangzhou, China) was incubated at 37 °C for 2 h, stained with DAB, and then stained with hematoxylin (Solarbio, Beijing, China) and sealed with a neutral rubber (Solarbio, Beijing, China). It was also observed and photographed under a microscope (Leica, Wetzlar, Germany). The immunohistochemical results were statistically analyzed using image Pro plus (V6.0) software. The statistical results show the percentage of positively stained tumor cells to all tumor cells (% Area).

### 4.12. Acute Toxicity Analysis

The toxicity of kzl052 in vivo was tested by an acute toxicity analysis in male BALB/c mice. A total of 20 mice were divided into 5 groups randomly. Each group was given a one-time treatment of kzl052 concentrations at 0 mg/kg, 50 mg/kg, 100 mg/kg, 200 mg/kg, and 500 mg/kg. The survival or death status of mice within 24 h was observed and recorded. The survival and death rate of mice were calculated.

### 4.13. Statistical Analysis

All data are expressed as mean ± standard deviation. Single factor analysis of variance was performed using GraphPad Prism 6 software with * *p* < 0.05, ** *p* < 0.01, and *** *p* < 0.001.

## Figures and Tables

**Figure 1 ijms-26-06093-f001:**
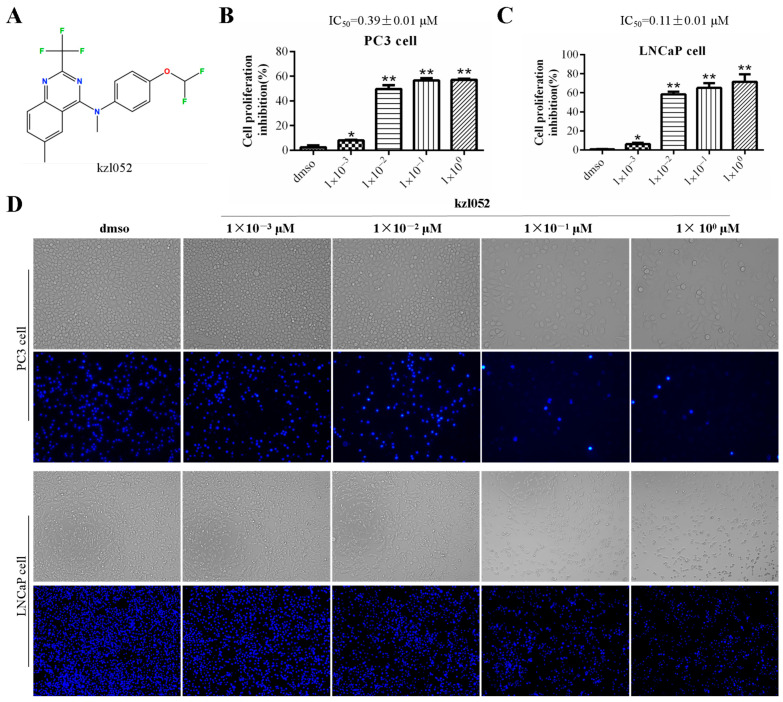
kzl052 inhibits the cell growth of PCa. (**A**) The structure of compound kzl052. (**B**,**C**) MTT results of PC3 and LNCaP cells treated with different concentrations of kzl052 after 24 h. (**D**) Bright field and Hoechst staining results of PC3 and LNCaP cells treated with kzl052 at different concentrations after 24 h. * *p* < 0.05, ** *p* < 0.01, and all experiments were repeated three times.

**Figure 2 ijms-26-06093-f002:**
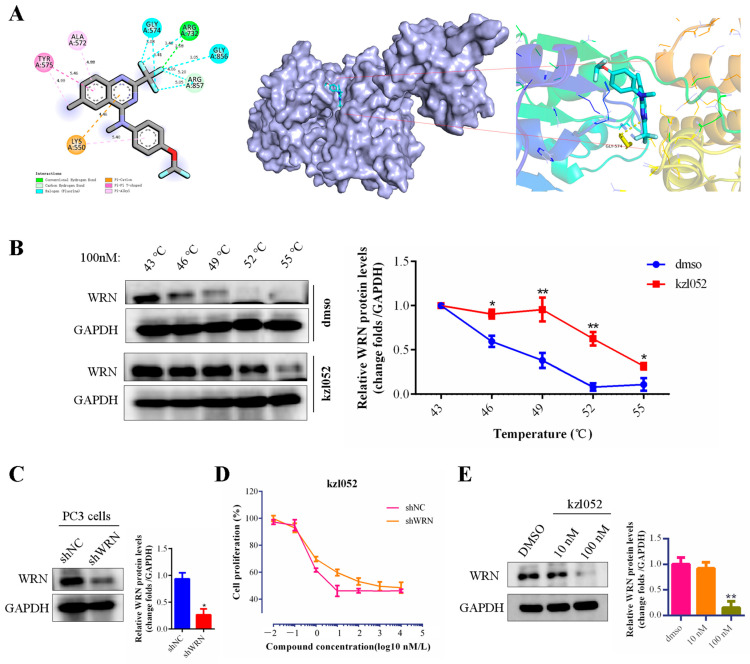
kzl052 inhibited the PCa cell growth by targeting WRN. (**A**) Visual results of molecular docking between kzl052 and WRN (PDB:6YHR). (**B**) CETSA results of kzl052 combined with WRN protein. (**C**) Western blot results of WRN-silenced PC3 cells. (**D**) Cell growth curve of kzl052 in WRN-silenced cells. (**E**) The WRN protein levels under kzl052 treatment for 24 h. * *p* < 0.05, ** *p* < 0.01, and all experiments were repeated three times.

**Figure 3 ijms-26-06093-f003:**
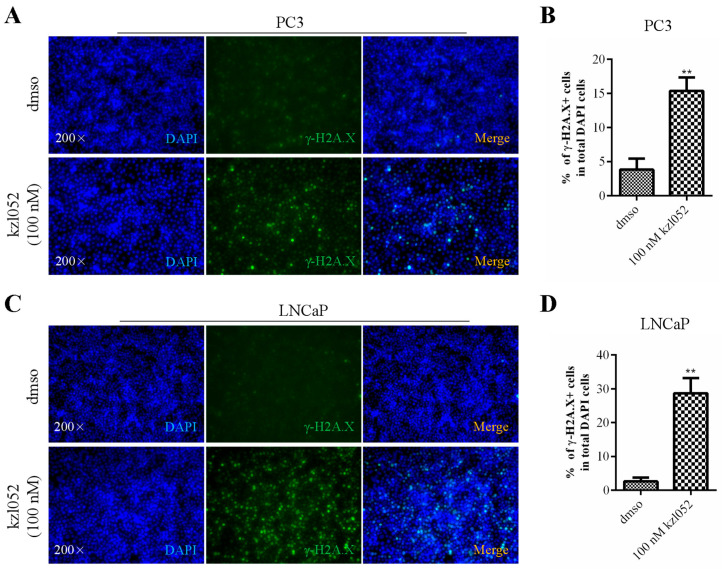
kzl052 promotes DNA damage in PC3 and LNCaP cells. γ-H2A.X immunofluorescence staining and its statistical results in PC3 (**A**,**B**) and LNCaP (**C**,**D**) cells. ** *p* < 0.01, and all experiments were repeated three times.

**Figure 4 ijms-26-06093-f004:**
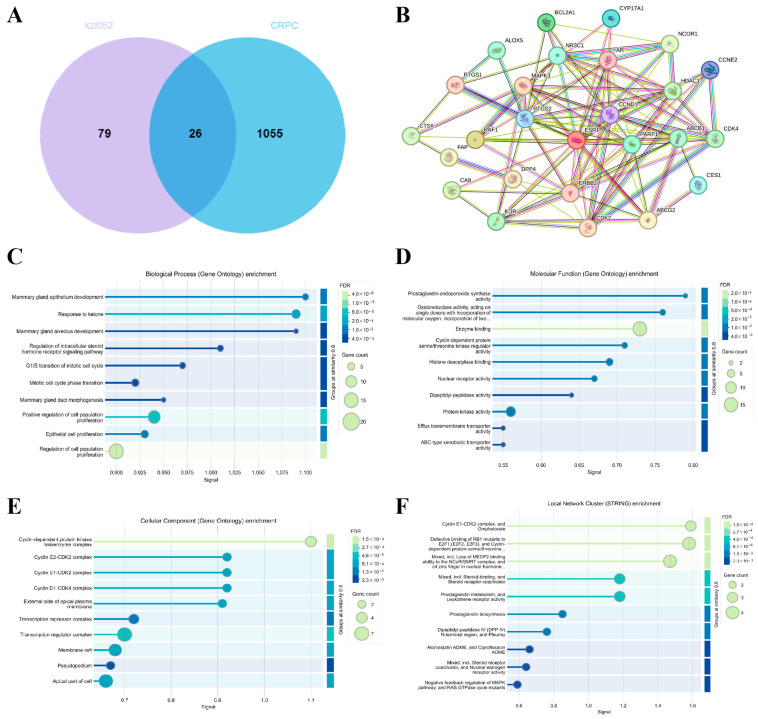
Regulatory signal pathway of kzl052 against PCa. (**A**) Venny analysis of kzl052 target genes and CRPC key genes. (**B**) Results of protein–protein interactions (PPI) of key genes. (**C**–**E**) Gene ontology function annotation results of key genes. (**F**) The pathway analysis results of target genes.

**Figure 5 ijms-26-06093-f005:**
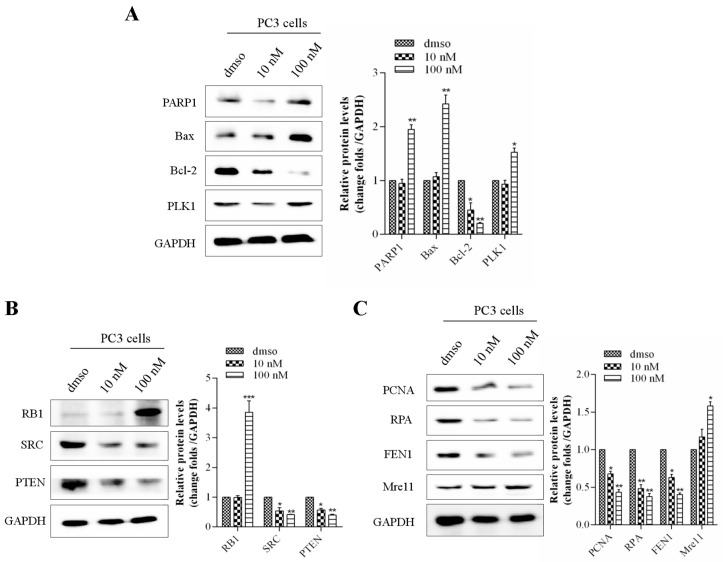
kzl052 regulates the expression of the replication fork, key protein components, and the RB signaling pathway. (**A**) kzl052 regulates the expression of key factors of the cell cycle and cell apoptosis. (**B**) kzl052 regulates RB signals. (**C**) kzl052 regulates the expression of key factors of the replication fork complexes. * *p* < 0.05, ** *p* < 0.01, *** *p* < 0.001, and all experiments were repeated three times.

**Figure 6 ijms-26-06093-f006:**
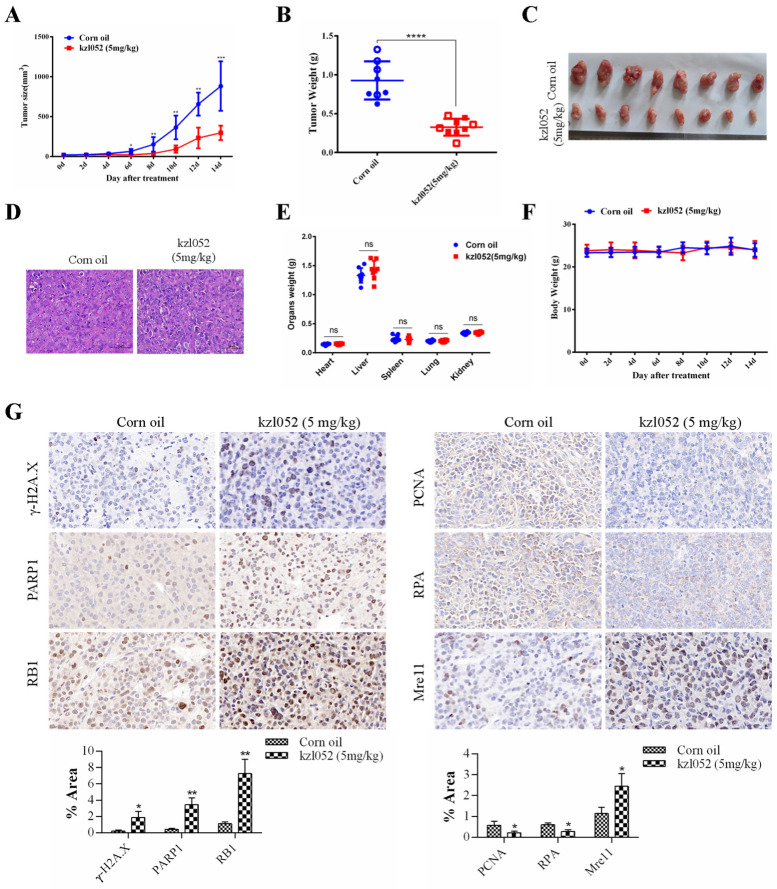
kzl052 inhibited PCa cell growth by regulating DNA replication fork stability in vivo. (**A**) Tumor volume changes after treatment with kzl052. (**B**) Tumor quality. (**C**) Tumor tissue. (**D**) H&E staining results. (**E**) Mouse body weight. (**F**) Internal organ morphology in mice. (**G**) The immunohistochemistry and its statistical results of γ-H2A.X, PARP1, RB1, PCNA, RPA, and Mre11 in cancer tissues. * *p* < 0.05, ** *p* < 0.01, and all experiments were repeated three times. * *p* < 0.05, ** *p* < 0.01, *** *p* < 0.001, **** *p* < 0.0001; the number of mice in the animals was eight.

**Figure 7 ijms-26-06093-f007:**
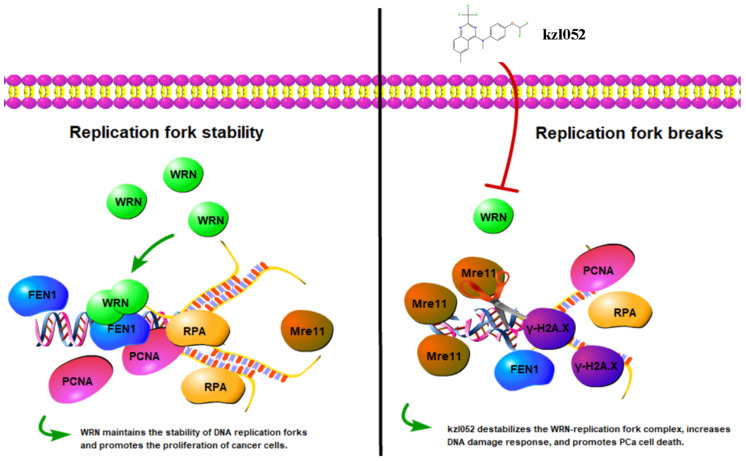
kzl052 against PCa by regulating WRN—DNA replication fork stability.

**Table 1 ijms-26-06093-t001:** The IC50 values of kzl052 against normal cell lines.

Cell Lines	IC_50_ Values (μM)		
	kzl052	Docetaxel	NSC 617145
LX2	3.12 ± 0.09 *	0.41 ± 0.11	1.81 ± 0.27
HK2	0.92 ± 0.24 ***	0.11 ± 0.01	5.64 ± 0.96

Data are expressed as mean ± SD, n = 3, * *p* < 0.05, *** *p* < 0.001. NSC 617145 were used as a positive control, while normal human liver cells (LX2) and normal human renal epithelial cells (HK2) were used to evaluate the safety of kzl052.

**Table 2 ijms-26-06093-t002:** Acute toxicity assay of kzl052 in BALB/c mice.

Concentration	Number	Volume	Survival Number	Death Number	Survival Rate	Mortality Rate
0 mg/kg	4	100 μL	4	0	100%	0
50 mg/kg	4	100 μL	4	0	100%	0
100 mg/kg	4	100 μL	4	0	100%	0
200 mg/kg	4	100 μL	4	0	100%	0
500 mg/kg	4	100 μL	1	3	25%	75%

## Data Availability

Data will be made available on request.

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
