# Peer review of "Quinazoline Derivative kzl052 Suppresses Prostate Cancer by Targeting WRN Helicase to Stabilize DNA Replication Forks"

_ijms, 2025, doi:10.3390/ijms26136093_

Round 1

Reviewer 1 Report

Comments and Suggestions for Authors

This study investigates the mechanism by which the quinazoline derivative kzl052 inhibits prostate cancer (PCa) through targeting WRN helicase, demonstrating its antitumor effects by disrupting DNA replication fork stability and modulating the RB pathway. The research combines molecular docking, in vitro/in vivo experiments, and bioinformatics analysis to confirm the WRN-dependent activity of kzl052 and its low in vivo toxicity, providing a novel strategy for WRN-targeted therapy. However, the following modifications are recommended:

  1. Add a schematic diagram summarizing the molecular mechanism of kzl052 in regulating replication fork and RB pathway.
  2. Include analysis of the correlation between WRN expression and prostate cancer stages.
  3. Supplement drug toxicity data.
  4. Compare with existing WRN inhibitors (e.g., HRO761) to analyze the potential clinical advantages of kzl052.
  5. Specify cell passage numbers, antibody catalog numbers, animal housing conditions, and randomization methods.
  6. Provide evidence for triplicate experimental repeats.
  7. Include original images of animal experiments.
  8. The loading control GAPDH appears uneven in Figures 2B and 2C.
  9. Some figures are blurry, such as Figure 2A.

Author Response

Review #1

Comments and Suggestions for Authors

This study investigates the mechanism by which the quinazoline derivative kzl052 inhibits prostate cancer (PCa) through targeting WRN helicase, demonstrating its antitumor effects by disrupting DNA replication fork stability and modulating the RB pathway. The research combines molecular docking, in vitro/in vivo experiments, and bioinformatics analysis to confirm the WRN-dependent activity of kzl052 and its low in vivo toxicity, providing a novel strategy for WRN-targeted therapy. However, the following modifications are recommended:

  1. Add a schematic diagram summarizing the molecular mechanism of kzl052 in regulating replication fork and RB pathway.

Response: Thank you very much for your suggestion. We already added a schematic diagram summarizing the molecular mechanism of kzl052 in regulating replication fork and RB pathway at Figure 7.

2.Include analysis of the correlation between WRN expression and prostate cancer stages. 

Response: Thank you very much for your suggestion. The analysis results of the UALCAN database showed that there was no significant difference between WRN expression and the death of PCa patients, which might be related to the smaller sample size. However, WRN has a certain relationship with the staging and progression of prostate cancer. The results are shown in Figure S1.

3.Supplement drug toxicity data. 

Response: Thank you very much for your suggestion. We have supplemented the normal cytotoxicity test and the acute toxicity test in Balb/C mice. The results are shown in Table 1 and Table 2.

4.Compare with existing WRN inhibitors (e.g., HRO761) to analyze the potential clinical advantages of kzl052. 

Response: Thank you very much for your suggestion. In this study, kzl052 did not affect the helicase activity of WRN (Figure S1), and its anti-cancer sensitivity might be related to its non-enzymatic mechanism [1]. Our results confirmed that kzl052 might affect the function of WRN by regulating the binding of WRN to the DNA replication fork complex. However, the currently reported WRN inhibitors have no significant effect on non-MSI-H tumors. This could be the advantage of kzl052 in non-BRCA-deficient prostate cancer. This study can expand the coverage of new WRN inhibitors for non-MSI-H type tumors and has potential clinical advantages.

[1] Gupta, P., Majumdar, A. G., & Patro, B. S. (2022). Non-enzymatic function of WRN RECQL helicase regulates removal of topoisomerase-I-DNA covalent complexes and triggers NF-κB signaling in cancer. Aging cell, 21(6), e13625.

5.Specify cell passage numbers, antibody catalog numbers, animal housing conditions, and randomization methods. 

Response: Thank you very much for your suggestion. We have added relevant descriptions in the method section, and all modifications have been marked in text.

6.Provide evidence for triplicate experimental repeats. 

Response: Thank you very much for your suggestion.We have added the statistical results of the repeated experiments in Figure 3 and Figure 5.

7.Include original images of animal experiments.

Response: Thank you very much for your suggestion. We have provided the original images of the animal experiments. Please refer to them in original images fig6E.

8.The loading control GAPDH appears uneven in Figures 2B and 2C.

Response: Thank you very much for your suggestion. Although the internal parameters in Figures 2B are not neat enough, there are statistical results, which do not affect its conclusion. For Figures 2C, the relevant data of WRN knockdown cell lines have been published [1-2]. In these studies, we conducted sensitivity detection and analysis of relevant small molecule compounds on the constructed WRN knockdown cell lines.

[1] Yu J#, Zhou Y#, Liang G, Cheng S, Wei J, Li H, Liu X, You C, Mao M, Ahmad M, Yu G*, Xu B*, Luo H*. Quinazoline derivatives inhibit cell growth of prostate cancer as a WRN helicase dependent manner by regulating DNA damage repair and microsatellite instability. Bioorganic chemistry. 2024, 153:107963.

[2] Yu G#, Yu J#, Zhou Y, Liu K, Peng X, Xu G, Chen C, Meng X, Zeng X, Wu H, Zan N, Luo H, Xu B. Discovery of novel quinazoline derivatives containing trifluoromethyl against cell proliferation by targeting werner helicase. Molecular diversity. 2025 Mar 28.

9.Some figures are blurry, such as Figure 2A.

Response: Thank you very much for your suggestion. We have updated the figure 2A to ensure sufficient clarity.

Reviewer 2 Report

Comments and Suggestions for Authors

This manuscript by Yu Jia and co-workers describes the anti-proliferative effects of a novel quinazoline on prostate cancer (PCa) cells via targeting the WRN helicase and destabilizing DNA replication forks. The work includes extensive in vitro data and also a study with mice. Targeting WRN in PCa is timely and can expand therapeutic options.

I don’t see an immediate connection between kzl052 and WRN. It is clear that kzl052 kills tumoural cells, though no control with non-tumoural cells is present, but I don’t see a clear description or explanation of what is happening. Looking into fig 2B, it appears that DMSO alone leads to lower WRN amounts than kzl052, but fig 2C suggests the opposite. Also, neither figure shows WRN protein levels in non treated cells. This needs to be included or at least discussed, and the apparent contradiction of figs 2B and 2C needs to be resolved.

Further below, a claim that the compounds binds WRN is made, but how would this lead to lower WRN levels? Is it because it promotes apoptosis and there’s overall less living cells? This needs to be clarified, particularly in mechanistic terms.
The authors propose that kzl052 “binds WRN,” but simultaneously report reduced WRN levels. Binding generally stabilizes target proteins in CETSA. How does kzl052 both appear to stabilize WRN and reduce its abundance? The authors should consider potential mechanisms for this:
proteasomal degradation - binding could induce conformational changes targeting WRN for degradation? 
Is it transcriptional regulation?
Is it an indirect apoptosis effect as I mentioned above?
I believe at least one assay to pinpoint whether WRN lowering is a direct of a secondary results of cell death should be included. These and other mechanistic possibilities should then be discussed.

Also, all WRN measurements were made only in cancer cell lines treated, with kzl052 or DMSO. Without results from non-tumoural prostate cells, it’s impossible to tell whether kzl052’s effects on WRN are cancer-specific or general cytotoxicity.

Also, methods needs to be detailed all across the board. For example, MTT assay, docking parameters, bioinformatics tools and parameters, Blot protocols, animal housing and feeding conditions, among many others, need to detailed.

I’m recommending reconsideration after major revisions to give the authors opportunity to address some issues on mechanism and several aspects of the manuscript, and also beacuse the manuscript would gain from a couple of additional experiments. Language-wise, the manuscript needs to be thoroughly revised, as there are many typos, many sentence fragments, and inconsistent use of verbal tenses.

Specific comments

Abstract: Quantitative data should be included here. Values like IC50 and tumor inhibition % allow readers to quickly assess the potency and in vivo potential relevance of kzl052.

Introduction
the opening sentences indicate PCa is rising in Asia, but lacks quantitative incidence or mortality rates; including epidemiological data would give a clearer picture of disease burden in China, Japan, and India, vs. EU+USA
The introduction jumps from PCa (1st paragraph) to WRN (2nd paragraph) without a clear link. A brief explanation framing the limitations of current targeted therapies in PCa would better motivated the study of WRN as a novel target. 
The discussion of WRN polymorphisms (Leu1074Phe) and mutations (G327X) in PCa is valuable, but these should be linked to functional consequences - how to they alter WRN activity?

Results & Methods

Section 2.1
line 58: please define the PC3 and LNCap cell lines.
Section 4.2: please define the MTT dye solution - Brand? Reference? Concentration? Kit?
Section 4.2: what is an “enzymoscope”? I’m afraid is not a trivial name.
Figure 2.1: xx axis in B and C need units
This section shows that kzl052 kills tumoural cell lines. But no control on non-tumoural cells is shown. Please include equivalent MTT data using a non-tumoural cell line or another type of relevant and equivalent control.
- line 66: Hoechst, not hochest; occurs in other places

Section and 2.2 and 4.3
please specify the docking protocol - size and center of the box? What was considered rigid and what was considered flexible? What are the residues in the binding site?
How was the interaction analysis performed? Manually or using a specific software tool?
Binding energy alone does not provide a lot of information. Please include the binding energy obtained with the same approach for a relevant control or for a substrate model

Section 2.3
please explain exactly what DAPI staining shows, what gamm-H2AX shows, and how these results clearly support the DNA damage conclusion. It is not trivial.

Section 2.4 and 4.6

Figure 4:
How were the PPI determined? This needs to be explicit. 
C to E, also F: please indicate clearly how were these GO obtained (software? Parameters?), it is not clear from either here of the methods.
Line 107: I believe (E) should be (F).
    - 
Lines 213 to 216:
references for SwissTargetPrediction, Super-PRED database, GeneCards, etc are required.
How was the GO analysis performed?
How was the pathway analysis performed?

Figure 5 and section 2.5

“Western ot results showed that kzl052 significantly up-regulated the expression of lPARP1, Bax, PLK1 and RB1, inhibited the protein levels of Bcl-2, SRC and PTEN, and promoted apoptosis (Figure 5A-B).” The authors need to clearly state what is the role of these proteins, and how the results obtained sustain the apoptosis-inducing conclusion.
“In addition, kzl052 significantly inhibited the expression of PCNA, RPA and FEN1, and upregulated the protein level of Mre11, suggesting that kzl052 promoted the instability of DNA replication forks complex (Figure 5C).” - same comment: The authors need to clearly state what is the role of these proteins, and how the results obtained sustain the instability conclusion.
In these two aspects, the link between the observed results and the conclusions, highlighting the authors rationale, must be made crystal clear.

Author Response

Review #2

Comments and Suggestions for Authors

Comments and Suggestions for Authors

This manuscript by Yu Jia and co-workers describes the anti-proliferative effects of a novel quinazoline on prostate cancer (PCa) cells via targeting the WRN helicase and destabilizing DNA replication forks. The work includes extensive in vitro data and also a study with mice. Targeting WRN in PCa is timely and can expand therapeutic options.

I don’t see an immediate connection between kzl052 and WRN. It is clear that kzl052 kills tumoural cells, though no control with non-tumoural cells is present, but I don’t see a clear description or explanation of what is happening. Looking into fig 2B, it appears that DMSO alone leads to lower WRN amounts than kzl052, but fig 2C suggests the opposite. Also, neither figure shows WRN protein levels in non treated cells. This needs to be included or at least discussed, and the apparent contradiction of figs 2B and 2C needs to be resolved. 

Response: Thank you very much for your suggestion. We have supplemented the inhibitory activity against non-tumor cells. The results are shown in Table 1. Figure 2B and Figure 2C are two different experiments. Figure 2B is the Cellular Thermal Shift Assay, which discusses the stability of the binding complex of kzl052 and WRN proteins at different temperatures, and the results show that the binding of kzl052 and WRN proteins can enhance the temperature tolerance of the compound - protein complex. Figure 2C shows the western blotting detection of the WRN knockdown in PC3 cells. Furthermore, we provided the expression levels of WRN protein in different prostate cancer cell lines, and the results are shown in Figure S1. All the modifications have been marked in text.

Further below, a claim that the compounds binds WRN is made, but how would this lead to lower WRN levels? Is it because it promotes apoptosis and there’s overall less living cells? This needs to be clarified, particularly in mechanistic terms.

Response: Thank you very much for your suggestion. In this study, we did not detect the effect of kzl052 on the expression level of WRN. Therefore, there was no conclusion such as "leading to a decrease in WRN level" in the article. Figure 2B and Figure 2C are two different experiments. Figure 2B is the Cellular Thermal Shift Assay, which discusses the stability of the binding complex of kzl052 and WRN proteins at different temperatures, and the results show that the binding of kzl052 and WRN proteins can enhance the temperature tolerance of the compound - protein complex. Figure 2C shows the western blotting detection of the WRN knockdown in PC3 cells and the results showed that the expression of WRN in PC3 cells decreased significantly after transfection with pLV-shWRN.

The authors propose that kzl052 “binds WRN,” but simultaneously report reduced WRN levels. Binding generally stabilizes target proteins in CETSA. How does kzl052 both appear to stabilize WRN and reduce its abundance? The authors should consider potential mechanisms for this:

proteasomal degradation - binding could induce conformational changes targeting WRN for degradation?

Is it transcriptional regulation?

Is it an indirect apoptosis effect as I mentioned above?

I believe at least one assay to pinpoint whether WRN lowering is a direct of a secondary results of cell death should be included. These and other mechanistic possibilities should then be discussed.

Response: Thank you very much for your suggestion. In this study, we did not detect the effect of kzl052 on the expression level of WRN. Therefore, there was no conclusion such as "leading to a decrease in WRN level" in the article. Therefore, it does not involve "how kzl052 stabilizes WRN and reduces its abundance" and its regulatory mechanism. However, the comments of the review provide us with a new mechanism for the subsequent research on WRN helicase inhibitors, and we will explore proteasome degradation and binding, transcription regulation and other aspects. Furthermore, we also detected the effect of kzl052 on the expression of WRN protein. The results showed that kzl052 down-regulated the expression of WRN (shown in Figure 2E). The results of Figure 5 and Figure S1 suggested that kzl052 may affect the non-enzymatic function of WRN and play an anti-tumor role by disrupting the binding mechanism between WRN and the key factors of DNA replication forks.

Also, all WRN measurements were made only in cancer cell lines treated, with kzl052 or DMSO. Without results from non-tumoural prostate cells, it’s impossible to tell whether kzl052’s effects on WRN are cancer-specific or general cytotoxicity.

Response: Thank you very much for your suggestion. We have supplemented the proliferation inhibitory activity of kzl052 against human renal tubular epithelial cells HK2 and human normal hepatic stellate cells LX2. The IC50 values are shown in Table 1.

Also, methods needs to be detailed all across the board. For example, MTT assay, docking parameters, bioinformatics tools and parameters, Blot protocols, animal housing and feeding conditions, among many others, need to detailed.

Response: Thank you very much for your suggestion. We have provided a detailed description of the method section. All the modifications have been marked in text.

I’m recommending reconsideration after major revisions to give the authors opportunity to address some issues on mechanism and several aspects of the manuscript, and also beacuse the manuscript would gain from a couple of additional experiments. Language-wise, the manuscript needs to be thoroughly revised, as there are many typos, many sentence fragments, and inconsistent use of verbal tenses.

Response: Thank you very much for your suggestion. We have revised the manuscript for language, tense, and polished it using ChatGPT. If there is any improvement, please contact us in time.

Specific comments

Abstract: Quantitative data should be included here. Values like IC50 and tumor inhibition % allow readers to quickly assess the potency and in vivo potential relevance of kzl052. 

Response: Thank you very much for your suggestion. We have made revisions in the abstract section.

Introduction

the opening sentences indicate PCa is rising in Asia, but lacks quantitative incidence or mortality rates; including epidemiological data would give a clearer picture of disease burden in China, Japan, and India, vs. EU+USA

The introduction jumps from PCa (1st paragraph) to WRN (2nd paragraph) without a clear link. A brief explanation framing the limitations of current targeted therapies in PCa would better motivated the study of WRN as a novel target. 

The discussion of WRN polymorphisms (Leu1074Phe) and mutations (G327X) in PCa is valuable, but these should be linked to functional consequences - how to they alter WRN activity?

Response: Thank you very much for your suggestion. We have added epidemiologically relevant data such as PCa incidence or mortality in the introduction. We have added a connection between the first and second paragraphs, supplemented the limitations of PCa-targeted therapy and explained the significance of WRN as a new target research. In addition, the relevant descriptions of the effects of WRN polymorphisms and mutations on WRN activity have also been added.

Results & Methods

Section 2.1

line 58: please define the PC3 and LNCap cell lines.

Response: Thank you very much for your suggestion. We have defined the PC3 and LNCap cell lines, and the modifications have been marked in text.

Section 4.2: please define the MTT dye solution - Brand? Reference? Concentration? Kit? 

Response: Thank you very much for your suggestion.We have added the concentration and brand of MTT dye solution.

Section 4.2: what is an “enzymoscope”? I’m afraid is not a trivial name.

Response: Thank you very much for your suggestion. Sorry for such a mistake, we have corrected it.

Figure 2.1: xx axis in B and C need units 

Response: Thank you very much for your suggestion. The Y-axis of the statistical graphs in Figures 2B and 2C represents multiples of the expression of the target protein in units of change (folds/GAPDH).

This section shows that kzl052 kills tumoural cell lines. But no control on non-tumoural cells is shown. Please include equivalent MTT data using a non-tumoural cell line or another type of relevant and equivalent control. 

Response: Thank you very much for your suggestion. We have supplemented the proliferation inhibitory activity of kzl052 against human renal tubular epithelial cells HK2 and human normal hepatic stellate cells LX2. The IC50 values are shown in Table 1.

- line 66: Hoechst, not hochest; occurs in other places

Response: Thank you very much for your suggestion. Sorry for such a mistake, we have corrected it.

Section and 2.2 and 4.3

please specify the docking protocol - size and center of the box? What was considered rigid and what was considered flexible? What are the residues in the binding site?

How was the interaction analysis performed? Manually or using a specific software tool?

Binding energy alone does not provide a lot of information. Please include the binding energy obtained with the same approach for a relevant control or for a substrate model

Response: Thank you very much for your suggestion. We have added docking protocol in the section of method "4.4. AutoDocking", and described the box, interaction analysis and docking software tools.

Section 2.3

please explain exactly what DAPI staining shows, what gamm-H2AX shows, and how these results clearly support the DNA damage conclusion. It is not trivial.

Response: Thank you very much for your suggestion. We have added the relevant descriptions of the DAPI and γ-H2A.X staining results as well as the conclusions inferred from these results.

Section 2.4 and 4.6

Figure 4:

How were the PPI determined? This needs to be explicit. 

C to E, also F: please indicate clearly how were these GO obtained (software? Parameters?), it is not clear from either here of the methods.

Line 107: I believe (E) should be (F).

Response: Thank you very much for your suggestion. We have added relevant descriptions on how to determine PPI, GO and pathway analysis in the "4.6.Bioinformatics analysis" section of the method. Furthermore, Figure 4(E) has been modified to (F).

Lines 213 to 216:

references for SwissTargetPrediction, Super-PRED database, GeneCards, etc are required.

How was the GO analysis performed?

How was the pathway analysis performed?

Response: Thank you very much for your suggestion. We have added references to SwisStart Get Prediction, Super-PRED database, GeneCards, etc. in the "4.6.Bioinformatics analysis" section of the method. GO analysis and pathway analysis were performed by STRING website. All modifications have been marked in text.

Figure 5 and section 2.5

“Western ot results showed that kzl052 significantly up-regulated the expression of lPARP1, Bax, PLK1 and RB1, inhibited the protein levels of Bcl-2, SRC and PTEN, and promoted apoptosis (Figure 5A-B).” The authors need to clearly state what is the role of these proteins, and how the results obtained sustain the apoptosis-inducing conclusion.

“In addition, kzl052 significantly inhibited the expression of PCNA, RPA and FEN1, and upregulated the protein level of Mre11, suggesting that kzl052 promoted the instability of DNA replication forks complex (Figure 5C).” - same comment: The authors need to clearly state what is the role of these proteins, and how the results obtained sustain the instability conclusion. 

In these two aspects, the link between the observed results and the conclusions, highlighting the authors rationale, must be made crystal clear.

Response: Thank you very much for your suggestion. We have added the relevant descriptions of proteins such as PARP1, Bax, PLK1, RB1, Bcl-2, SRC, PTEN, as well as PCNA, RPA and FEN1 in the corresponding parts of the text to strengthen the connection between the results and the conclusions. All the modifications have been marked in text.

Reviewer 3 Report

Comments and Suggestions for Authors

The authors identified kzl052 as a novel WRN inhibitor that inhibits castration-refractory prostate cancer progression with less toxicity - in a mouse model, providing a potential new option to treat PCa tumors.

The authors did not address the question whether kzl052 promotes apoptosis in CRPC via synthetic lethality as only non-BRCA deficient or non-MSI high prostate cancer cells have been studied, although, synthetic lethality seems to be the major domain of WRN inhibitors.

An interesting result is that IC50 for kzl052 in PC3 is about 3-fold higher than IC50 in LNCaP cells: what are the reasons, quite different, expression levels of WRN? More efficacious in a presumably neuroendocrine (PSA negative) tumor?

Further studies have been carried out with the PC3 cell line. Why did you not perform all experiments with both cell lines to cover the diversity of CRPC?

Why no consecutive testing of kzl052 in BRCA deficient or MSI high prostate cancer cells? CRPC is an extremely heterogeneous disease including neuroendocrine tumors and WRN inhibition reduces viability of BRCA2-deficient cells, potentiates cytotoxicity of a poly (ADP)ribose polymerase (PARP) inhibitor and induces synthetic lethality in MSI positive cancers.

WRN may be used as an alternative or auxiliary target for PARP inhibitors: Please discuss the already used PARP inhibitors!

Are toxicity data available from other animal models?

CRPC key genes? Considering the heterogeneity of CRPC, what is the basis of the data analysis? Neuroendocrine tumors included?

WRN inhibitors have no significant effect on non-MSI-H tumors. What is the advantage of kzl052 in non-BRCA deficient prostate cancer?

‘In addition, the high toxicity and side effects caused by off-target action further limit the clinical research of existing WRN inhibitors.’ No literature given!

‘In this study, kzl052 significantly increased the expression of PARP1, which may have a synergistic role in promoting cell death with PARP inhibitors’: ‘could be’, as side effects are potentially worsened.

Please discuss more focused, based on your experimental data and additional data on LNCaP cells, the position of kzl052, that can be not used as WRN inhibitor to induce synthetic lethality in contrast to other WRN inhibitors and explain what could be the advantage for treating non-BRCA deficient or non-MSI high prostate cancer. The enhancement of PARP seems to be an adverse event also in non-BRCA deficient or non-MSI high prostate cancer cells !?

Author Response

Review #3

Comments and Suggestions for Authors

The authors identified kzl052 as a novel WRN inhibitor that inhibits castration-refractory prostate cancer progression with less toxicity - in a mouse model, providing a potential new option to treat PCa tumors.

The authors did not address the question whether kzl052 promotes apoptosis in CRPC via synthetic lethality as only non-BRCA deficient or non-MSI high prostate cancer cells have been studied, although, synthetic lethality seems to be the major domain of WRN inhibitors.

Response: Thank you very much for your suggestion. In the studies we have reported, a series of novel WRN compounds, including kzl052, cannot promote the apoptosis of CRPC cells through synthetic lethality[1-2], because the synthetic lethality of WRN inhibitors only targets MSI-H type tumors and is ineffective against MSS type. Therefore, in this study, we only discussed the effect on non-BRCA-deficient prostate cancer cells. Because in the previous functional studies, we found that knockdown of WRN in the CRPC cell line did not cause synthetic lethality. At the same time, we also found that simultaneous inhibition of other key genes with WRN could achieve synthetic lethality (data not shown). Therefore, in this study, we only discussed the effect of kzl052 on non-BRCA-deficient prostate cancer cells and did not explore the synthetic lethality.

[1] Yu G, Yu J, Zhou Y, Liu K, Peng X, Xu G, Chen C, Meng X, Zeng X, Wu H, Zan N, Luo H, Xu B. Discovery of novel quinazoline derivatives containing trifluoromethyl against cell proliferation by targeting werner helicase. Mol Divers. 2025 Mar 28.

[2] Yu J, Zhou Y, Liang G, Cheng S, Wei J, Li H, Liu X, You C, Mao M, Ahmad M, Yu G, Xu B, Luo H. Quinazoline derivatives inhibit cell growth of prostate cancer as a WRN helicase dependent manner by regulating DNA damage repair and microsatellite instability. Bioorg Chem. 2024 Dec;153:107963.

An interesting result is that IC50 for kzl052 in PC3 is about 3-fold higher than IC50 in LNCaP cells: what are the reasons, quite different, expression levels of WRN? More efficacious in a presumably neuroendocrine (PSA negative) tumor?

Response: Thank you very much for your suggestion. According to relevant research reports, in the PCa cell line, LNCAP cells belong to the MSI-H type [1], while PC3 cells belong to the MSS type [2]. We believe that kzl052, as a new type of WRN inhibitor, is more sensitive to MSI-H type cancer cells, which is consistent. As for whether it is related to neuroendocrine (PSA negative) tumors, it has not been discussed and is also a good idea to delve into.

[1] Yeh CC, Lee C, Dahiya R. DNA mismatch repair enzyme activity and gene expression in prostate cancer. Biochem Biophys Res Commun. 2001 Jul 13;285(2):409-13.

[2] Sagredou S, Dalezis P, Papadopoulou E, Voura M, Deligiorgi MV, Nikolaou M, Panayiotidis MI, Nasioulas G, Sarli V, Trafalis DT. Effects of a Novel Thiadiazole Derivative with High Anticancer Activity on Cancer Cell Immunogenic Markers: Mismatch Repair System, PD-L1 Expression, and Tumor Mutation Burden. Pharmaceutics. 2021 Jun 15;13(6):885.

Further studies have been carried out with the PC3 cell line. Why did you not perform all experiments with both cell lines to cover the diversity of CRPC?

Response: Thank you very much for your suggestion. Compared with BRCA-deficient PCa cell lines, there are fewer related studies on non-BRCA-deficient PC3 cells. We believe that the inhibitory mechanism of kzl052 on PC3 is more worthy of exploration. But this is a very good suggestion. We will cover all CRPC cell lines in the subsequent research.

Why no consecutive testing of kzl052 in BRCA deficient or MSI high prostate cancer cells? CRPC is an extremely heterogeneous disease including neuroendocrine tumors and WRN inhibition reduces viability of BRCA2-deficient cells, potentiates cytotoxicity of a poly (ADP)ribose polymerase (PARP) inhibitor and induces synthetic lethality in MSI positive cancers.

Response: Thank you very much for your suggestion. As described earlier, in the studies we have reported, a series of novel WRN compounds, including kzl052, were unable to promote the apoptosis of CRPC cells through synthetic lethality [1-2]. The proportion of non-BRCA-deficient cancers is much higher, so why not discuss the role of WRN inhibitors in non-BRCA-deficient cancers? Therefore, we only discussed the effect on non-BRCA-deficient prostate cancer cells in this study. Furthermore, in the previous functional studies, we found that knockdown of WRN in the CRPC cell line did not cause synthetic lethality. At the same time, we also discovered that simultaneous inhibition of other key genes with WRN could achieve synthetic lethality (data not shown). At present, we have discovered the lead compounds of the dual-target inhibitors (data not shown), and the synthetic lethality in non-BRCA-deficient cancers will be explored in the subsequent research.

[1] Yu G, Yu J, Zhou Y, Liu K, Peng X, Xu G, Chen C, Meng X, Zeng X, Wu H, Zan N, Luo H, Xu B. Discovery of novel quinazoline derivatives containing trifluoromethyl against cell proliferation by targeting werner helicase. Mol Divers. 2025 Mar 28.

[2] Yu J, Zhou Y, Liang G, Cheng S, Wei J, Li H, Liu X, You C, Mao M, Ahmad M, Yu G, Xu B, Luo H. Quinazoline derivatives inhibit cell growth of prostate cancer as a WRN helicase dependent manner by regulating DNA damage repair and microsatellite instability. Bioorg Chem. 2024 Dec;153:107963.

WRN may be used as an alternative or auxiliary target for PARP inhibitors: Please discuss the already used PARP inhibitors!

Response: Thank you very much for your suggestion. We have added the PARP inhibitor (olaparib) that has been used in the relevant reports.

Are toxicity data available from other animal models?

Response: Thank you very much for your suggestion. We have supplemented the acute toxicity experiments of kzl052 in Balb/c mice and the results are shown in Table 2.

CRPC key genes? Considering the heterogeneity of CRPC, what is the basis of the data analysis? Neuroendocrine tumors included?

Response: Thank you very much for your suggestion. The CRPC key genes selected in this study were from the GeneCards database (this data integrates data from more than 100 websites, making the data more comprehensive). No studies have reported the association between WRN or WRN inhibitors and the development of CRPC. Therefore, we selected more comprehensive key genes of CRPC to better explore the association between WRN and CRPC, but did not discuss neuroendocrine tumors.

WRN inhibitors have no significant effect on non-MSI-H tumors. What is the advantage of kzl052 in non-BRCA deficient prostate cancer?

Response: Thank you very much for your suggestion. kzl052 significantly promoted DNA damage and replication fork instability in non-BRCA deficient prostate cancer PC3 cells. However, the currently reported WRN inhibitors have no significant effect on non-MSI-H tumors. It could be the advantage of kzl052 in non-BRCA deficient prostate cancer. We have added relevant descriptions in the discussion section.

‘In addition, the high toxicity and side effects caused by off-target action further limit the clinical research of existing WRN inhibitors.’ No literature given!

Response: Thank you very much for your suggestion. We have deleted this sentence in the text.

‘In this study, kzl052 significantly increased the expression of PARP1, which may have a synergistic role in promoting cell death with PARP inhibitors’: ‘could be’, as side effects are potentially worsened.

Response: Thank you very much for your suggestion. We have modified the inappropriate description.

Please discuss more focused, based on your experimental data and additional data on LNCaP cells, the position of kzl052, that can be not used as WRN inhibitor to induce synthetic lethality in contrast to other WRN inhibitors and explain what could be the advantage for treating non-BRCA deficient or non-MSI high prostate cancer. The enhancement of PARP seems to be an adverse event also in non-BRCA deficient or non-MSI high prostate cancer cells !?

Response: Thank you very much for your suggestion. In this study, compared with other WRN inhibitors, kzl052 cannot be used as a WRN inhibitor to induce synthetic lethality. We think that it is may be related to its non-enzymatic mechanism of action [1]. Furthermore, kzl052 significantly promotes DNA damage and replication fork instability in non-BRCA-deficient prostate cancer PC3 cells. However, the currently reported WRN inhibitors have no significant effect on non-MSI-H tumors. This could be the advantage of kzl052 in non-BRCA-deficient prostate cancer. The upregulation of PARP1 protein caused by kzl052 is an adverse event in non-BRCA-deficient PCa cell lines, we believe that more evidence is needed to demonstrate it. We have added relevant descriptions in the text.

[1] Gupta, P., Majumdar, A. G., & Patro, B. S. (2022). Non-enzymatic function of WRN RECQL helicase regulates removal of topoisomerase-I-DNA covalent complexes and triggers NF-κB signaling in cancer. Aging cell, 21(6), e13625.

Round 2

Reviewer 2 Report

Comments and Suggestions for Authors

I would like to thank the authors for their careful answer. I consider the manuscript can be accepted as it is.

Author Response

Dear reviewers, Thank you very much!

Reviewer 3 Report

Comments and Suggestions for Authors

Thank you for the revisions!

Author Response

Dear reviewers, Thank you very much!